# Use of a Polymer Inclusion Membrane and a Chelating Resin for the Flow-Based Sequential Determination of Copper(II) and Zinc(II) in Natural Waters and Soil Leachates

**DOI:** 10.3390/molecules25215062

**Published:** 2020-10-31

**Authors:** Tânia C. F. Ribas, Charles F. Croft, M. Inês G. S. Almeida, Raquel B. R. Mesquita, Spas D. Kolev, António O. S. S. Rangel

**Affiliations:** 1CBQF—Centro de Biotecnologia e Química Fina—Laboratório Associado, Escola Superior de Biotecnologia, Universidade Católica Portuguesa, Rua Diogo Botelho 1327, 4169-005 Porto, Portugal; tpedro@porto.ucp.pt (T.C.F.R.); rmesquita@porto.ucp.pt (R.B.R.M.); 2School of Chemistry, University of Melbourne, Melbourne 3010, VIC, Australia; ccroft@student.unimelb.edu.au (C.F.C.); mariads@unimelb.edu.au (M.I.G.S.A.); s.kolev@unimelb.edu.au (S.D.K.)

**Keywords:** bi-parametric method, sequential injection analysis, micronutrients, polymer inclusion membrane, Chelex 100

## Abstract

A bi-parametric sequential injection method for the determination of copper(II) and zinc(II) when present together in aqueous samples was developed. This was achieved by using a non-specific colorimetric reagent (4-(2-pyridylazo)resorcinol, PAR) together with two ion-exchange polymeric materials to discriminate between the two metal ions. A polymer inclusion membrane (PIM) and a chelating resin (Chelex 100) were the chosen materials to retain zinc(II) and copper(II), respectively. The influence of the flow system parameters, such as composition of the reagent solutions, flow rates and standard/sample volume, on the method sensitivity were studied. The interference of several common metal ions was assessed, and no significant interferences were observed (<10% signal deviation). The limits of detection were 3.1 and 5.6 µg L^−1^ for copper(II) and zinc(II), respectively; the dynamic working range was from 10 to 40 µg L^−1^ for both analytes. The newly developed sequential injection analysis (SIA) system was applied to natural waters and soil leachates, and the results were in agreement with those obtained with the reference procedure.

## 1. Introduction

Sample preparation is considered an essential part of the analytical process. Some of the most commonly used sample pre-treatment methods are extraction techniques, namely solid-phase extraction (SPE), liquid–liquid extraction (LLE) and, more recently, membrane-based extraction techniques, among others [1].

The basic principles of both SPE and LLE can involve adsorption, partition or ion-exchange of solutes between the two different phases. In SPE, this occurs between a liquid phase (i.e., aqueous sample) and a solid phase (e.g., sorbent material), while, in LLE, this occurs between two immiscible liquid phases [2,3,4,5]. In recent years, these techniques have been evolving towards their miniaturization (i.e., solid-phase/liquid-phase microextraction), and different types of membranes have also been used as a support for organic phase in the microextraction process [1].

Separation techniques using solid resins or liquid membrane-based materials offer good selectivity, ease of operation, low use (or no use at all) of organic solvents, low cost, fast rates of analyte extraction and the possibility of being reused [2,3,6]. These factors can be even more advantageous when these techniques are conducted in an on-line fashion as part of flow analysis methods. Therefore, these on-line separation techniques have gained high interest in the last decade [3,7].

In the present study, the strategy was to explore different polymeric materials to selectively separate copper(II) and zinc(II) and subsequently use a color reagent able to produce a similar sensitivity to the determination of both metal ions. The chromogenic reagent selected was 4-(2-pyridylazo)resorcinol (PAR). PAR is a very commonly used chromogenic chelator for the spectrophotometric determination of various metal ions [8]. This reagent was selected because it is water soluble and does not need the use of organic solvents in the preparation of its solutions, unlike some chromogenic chelating agents used for metals quantification [9].

According to previous work [9], Chelex 100 efficiently retains copper at pH of 2. Thus, Chelex 100 was the chosen resin to avoid copper(II) interference in the zinc(II) determination. Chelex 100 resin is a styrene divinylbenzene copolymer, weakly acidic due to its carboxylic acid groups, thus allowing cation exchange. This sorbent material acts as chelating resin to bind metal ions and its selectivity is closely related to the pH of the chelating process. Advantage was taken from this property to retain the target analyte.

For the selective detection of zinc(II), a polymer inclusion membrane (PIM) was used [6,10]. PIMs are considered as a type of liquid membranes which have attracted considerable attention in recent years [11]. These membranes are usually fabricated by casting a solution containing an extractant, a base polymer, a plasticizer (if necessary) and a volatile solvent which dissolves all PIM components. After casting and evaporation of the volatile solvent a thin, flexible and stable polymeric film is formed. PIMs have the ability to selectively separate a species of interest depending on the extractant used [6]. Kolev et al. reported a PVC-based PIM with di-(2-ethylhexyl)phosphoric acid (D2EHPA) as the extractant, capable of extracting Zn(II) selectively [12]. This PIM composition was thus chosen for the zinc(II) extraction from the sample matrix, thus enabling the copper(II) determination.

The target analytes, copper(II) and zinc(II), are important micronutrients essential for the proper functioning of living organisms, but both become toxic in high concentrations. The presence of these metal ions in ground and surface water are a direct result of using soil fertilizers or of other anthropogenic activities influencing water quality. In this scenario, it is important to monitor these metal ions in natural waters, as they act as pollution indicators. Some research has already been done for the determination of these two metal ions based on flow systems [9,13,14,15,16,17,18,19]. However, to accomplish both determinations with a single sequential injection analysis (SIA) system, a mathematical discrimination treatment of the experimental results had to be used [9,14,15,16,17,18], except for the system developed by Santos et al. [13]. A different approach is here proposed which involves the use of two on-line columns containing different ion-exchange polymeric materials for the bi-parametric determination of copper(II) and zinc(II) in natural waters and soil leachates.

## 2. Results and Discussion

### 2.1. Preliminary Studies

Two different colorimetric reagents for metal ions, namely PAR and 1-(2-pyridylazo)-2-naphthol (PAN), were initially studied via wet/bench chemistry, to determine which would be more advantageous for the spectrophotometric quantification of both Cu(II) and Zn(II). Using the same conditions for both reagents (1 mL of 0.1 mmol L^−1^ of reagent solution, 1 mL of 0.5 µg L^−1^ of metal solution and 1 mL of 0.6 mmol L^−1^ of carbonate buffer solution at pH 10), a spectrum for each metal–reagent complex was obtained (Appendix A). By assessing the wavelength of maximum absorption of both metal complexes, it was observed that the signal was higher for the metal–PAR complexes, therefore PAR was chosen as the metal indicator to develop the SIA method. Additionally, PAR is water-soluble, which makes it easier to use.

### 2.2. Development of the SIA System

The development of the SIA system involved a number of optimization studies to assess the influence of some chemical and physical variables on the system’s analytical performance. As both complexes of PAR, with copper(II) or zinc(II) showed similar sensitivity under the same conditions (concentration of PAR, metal and buffer), copper(II) was chosen as the model analyte to conduct the optimization of the colorimetric reaction in the SIA system. The parameters assessed were the volumes of the PAR, sample and buffer solutions; the pH of the buffer solution; the reaction coil length; and the concentration of the PAR solution. These parameters were optimized in order to attain the highest sensitivity (calibration curve slope), the lowest reagent consumption and the most effective sampling rate.

#### 2.2.1. Study of the Reaction Conditions

The first study carried out was on the PAR, sample and buffer solution volumes. Different volumes of PAR (250–350 µL), buffer (10–30 µL) and sample/standard (300–650 µL) solutions were studied to evaluate their impact on the calibration curve parameters (Figure 1). The increase in the reagent volume resulted in an increase of the intercept (≈15%) but no increase of sensitivity (calibration curve slope); thus, the lowest volume was selected to ensure minimal reagent consumption. The sensitivity increased with increasing the buffer volume up to 20 µL, with almost no variation in the intercept (<5% variation). A similar behavior was observed for the sample volume, as the sensitivity increased with the increase in the sample volume up to 550 µL, but in this case the intercept decreased. For higher sample volumes, the slope decreased slightly (≈10% variation) and the intercept increased slightly (≈10% variation).

The chosen volumes were 250 µL for PAR, 20 µL for the buffer and 550 µL for the sample/standard solutions, as this combination displayed the highest slope and lowest intercept values for the calibration curve, indicating better reaction sensitivity with potentially lower detection limits for both determinations.

The reaction PTFE coil length was initially kept to a minimum (10 cm) allowing to physically connect the central port of the selection valve to the flow cell. As there was no significant difference (<10%) in the absorbance signal when the coil length was increased to 20 cm, the 10-cm length was used in the remaining experiments.

The influence of the buffer solution composition and pH was studied by comparing the sensitivity obtained when using a boric acid buffer (0.5 mol L^−1^) or a carbonate buffer (0.6 mol L^−1^). Both buffer solutions were tested at two different pH values: 10 or 11. No significant differences were observed (<5% sensitivity) between the boric acid buffer at pH 10 and 11 and the carbonate buffers at pH 10. However, with the use of the carbonate buffer, air bubbles were formed inside the tubing of the flow system. Additionally, when the carbonate buffer at pH 11 was used, poor repeatability of the signal was obtained. Hence, the chosen buffer solution was the boric acid solution at pH 11.

The influence of the PAR reagent concentration was also evaluated and the concentration of 25 µmol L^−1^ was chosen from the tested values (10, 25, 50 and 100 µmol L^−1^). When the concentration was increased above 25 µmol L^−1^, no significant variation (<10%) of the calibration curve slope was observed; the chosen concentration also produced a lower intercept value. 

#### 2.2.2. Study of the Retention of Copper(II) and Zinc(II).

Since PAR is sensitive to both copper(II) and zinc(II), a dual-extraction approach was adopted in order to be able to determine both metal ions individually and sequentially. The strategy chosen for the selective copper(II) determination consisted of the use of a PVC-based PIM containing D2EHPA as the extractant to retain zinc(II). Some studies were conducted to maximize the on-line retention of zinc(II) in natural water samples into the PIM and consequently determine copper(II). As the PIM column was linked to one of the peripheral ports of the selection valve, direct aspiration of a standard, containing both copper(II) and zinc(II), through the column was attempted first; however, the retention was not efficient. Alternatively, with the aim to enhance the interaction between the solution and the PIM, the standard was aspirated from another port of the selection valve to the holding coil of the SIA system, subsequently propelled through the column and aspirated back to the holding coil. Using this approach, two experiments were conducted. In the first one, the standard was propelled through the PIM column, followed by the sequential aspiration of PAR reagent, buffer solution and Zn(II)-free standard into the holding coil. After flow reversal, the stack of solution zones in the holding coil were propelled towards the detector. In the second experiment, a similar procedure was adopted except that the flow was stopped for 5 s when the standard zone was in the PIM column. No significant difference in the maximum absorbance signals was observed between the two experiments; thus, to further improve the retention of zinc(II), the procedure involving propelling and aspirating the standard zone through the PIM column was conducted twice (Appendix A). This retention procedure was thus the chosen approach for further optimization.

According to Paluch et al. [9], copper(II) can be retained in a column packed with Chelex 100 resin at pH of 2. Hence, for the selective zinc(II) determination, this was the strategy chosen to eliminate the interference of copper(II). A packed column was linked to one of the peripheral ports of the selection valve, through which the standard/sample was directly aspirated towards the holding coil, thus retaining any copper(II) present in the original standard/sample (Appendix A).

### 2.3. Interference Studies

Being a non-specific chromogenic reagent, PAR forms an orange complex with a variety of different metal ions and, thus, the potential interference from other metal ions was assessed. The ions that can be present in natural water samples and thus interfere with the proposed analytical method are described in Table 1. The selected concentrations for each ion corresponded to the maximum concentration that can be expected in environmental waters [20]. The obtained absorbance of a standard with and without the possible interfering ion was measured and the interference percentage calculated (Table 1).

The only interferences above 10% were from iron(III) and magnesium(II); however, values above 400 µg L^−1^ Fe(III) are not usually found in environmental waters, and the tested magnesium concentration was above the expected values. The interference from iron(III) has been previous reported and could be eliminated by precipitation with phosphate prior to the analysis [12].

### 2.4. Features

The features of the newly developed SIA method for the bi-parametric determination of copper(II) and zinc(II) are summarized in Table 2.

The limits of detection were calculated according to the IUPAC recommendations as the concentration corresponding to the sum of three times the standard deviation to the mean value of ten consecutive blank solution measurements [21,22].

The relative standard deviation (RSD) for Cu(II) and Zn(II) determination was calculated with twelve replicate analysis (four consecutive cycles) of a standard with 20 µg L^−1^ of each metal ion.

A complete cycle, which includes three replicates for each determination and the washing of the PIM column at the end, has the duration of 10 min. The corresponding PAR, sodium hydroxide and boric acid consumption per cycle is 8.1 µg, 1.4 mg and 5.6 mg, respectively.

### 2.5. Application to Natural Water and Soil Leachate Samples—Validation of the Method

The newly developed SIA system for the determination of copper(II) and zinc(II) was applied to river water samples (S1–S9) and soil leachates (S10–S14). The validation was attained by comparison of the results obtained with the newly developed SIA method with those obtained by the reference procedure (ICP-OES) (Table 3).

A linear relationship was established between the copper(II) and zinc(II) concentrations determined by the newly developed SIA system (C_SIA_ (µg L^−1^)) and the reference procedure (C_ICP_ (µg L^−1^)) (Appendix A). The linear regression for the copper(II) determination was C_SIA_ = 1.05 (±0.10) and C_ICP_ = 2.63 (±10.99), where the values in brackets represent the 95% confidence interval.

The linear regression for the zinc(II) determination was C_SIA_ = 1.03 (±0.02) and C_ICP_ = 0.46 (±2.53), where the values in brackets represent the 95% confidence interval. These data show that the estimated slope and intercept do not differ statistically from 1 and 0, respectively [23]. In addition, the relative deviation between the two sets of results proved that there were no significant differences between the newly developed SIA method and the reference procedure, RD ≤ 10% (Table 3).

The accuracy of the newly developed SIA method was evaluated by analyzing a certified reference water sample (ERM CA011—hard drinking water—metals) with 1963 ± 62 and 605 ± 17 µg L^−1^ of copper(II) and zinc(II), respectively. The concentration values obtained with the newly developed SIA system were 1789 ± 61 and 609 ± 24 µg L^−1^ for copper(II) and zinc(II), respectively, corresponding to relative deviations of −8.9% and +0.7%. These results indicate that the newly developed SIA system offered acceptable accuracy. 

## 3. Materials and Methods 

### 3.1. Reagents and Solutions

All solutions were prepared with analytical grade chemicals and MilliQ water (resistivity > 18 MΩcm, Millipore, Burlington, MA, USA).

A stock solution of 50.0 mg L^−1^ of copper(II) and zinc(II) were prepared by dilution of the respective 1000 mg L^−1^ atomic absorption standard solutions (Spectrosol, Poole, England). An intermediate solution of 500 µg L^−1^ of each metal solution was prepared by dilution of a 50.0 mg L^−1^ stock solution. Working standards, in the range 10–40 µg L^−1^ in 0.01 mol L^−1^ of nitric acid, were prepared weekly by dilution of a 500 µg L^−1^ intermediate solution with a 0.01 mol L^−1^ nitric acid solution.

A 0.01 mol L^−1^ nitric acid solution was prepared by dilution of the commercial concentrated nitric acid solution (d = 1.39; 65%, Merck; Darmstadt, Germany).

A buffer solution of 0.50 mol L^−1^ boric acid was prepared by dissolution of the solid (H_3_BO_3_, Aldrich, Germany) in a solution of 0.2 mol L^−1^ NaOH (Panreac, Chicago, IL, USA), with the final pH adjusted to 11.0 with a sodium hydroxide solution.

A 2 mmol L^−1^ stock solution of PAR (C_11_H_8_N_3_NaO_2_·H_2_O, Sigma-Aldrich, St Louis, MO, USA) was prepared by dissolving the corresponding quantity of the monosodium salt hydrate in water. A PAR reagent solution of 25 µmol L^−1^ was prepared weekly by dilution of the stock solution with MilliQ water. 

### 3.2. Preparation of the PIM Column

PIMs were produced by dissolving a mixture of 8.25 g of PVC and 6.75 g of D2EHPA in 165 mL of tetrahydrofuran (THF). Approximately 2.75 mL of this solution was cast into a glass ring with a 76-mm diameter which was positioned on a flat glass plate. All rings were covered with filter paper, a glass plate and a foil tray to control the evaporation of THF that was completed within a period of 48–72 h. The resulting PIM composition was 45 wt % D2EHPA and 55 wt % PVC.

PIMs were subsequently cut into strips of approximately 2 mm in width. A laboratory made column (5.5 cm length of Versilon 2001 tubing with 4.8 mm i.d.) was packed with the PIM stripes (approximately 100 mg) between two female Luer Tefzel connectors (P-624; Thermo Scientific, Waltham, MA, USA). The column was connected to one of the ports of the selection valve of the SIA system utilized in this study and subsequently used for zinc(II) retention.

Two columns were prepared and used along the method development and application to the sample analyses. This occurred for about four months for each column, without noticing significant deterioration on their performance.

### 3.3. Preparation of the Chelex 100 Column

A laboratory made column with 25 mm in length of Tygon tubing (Gilson, Villiers-le-Bel, France) with 1.85 mm i.d. and 67 µL inner volume was used to pack the chelating resin. Approximately 75 mg of Chelex 100 (mesh 200–400, Bio-Rad, Hercules, CA, USA), previously suspended in water, was introduced into the column between two pieces of dishwashing sponge. The column was connected to one of the ports of the SIA selection valve and used for copper retention.

One column was prepared and used throughout the entire method development, corresponding to approximately eight months, displaying a consistent performance.

### 3.4. Apparatus

Solutions were propelled in the experimental SIA system (Figure 2) by a syringe pump with a 5 mL barrel (Crison, Barcelona, Spain). The pump was connected to the central channel of a ten-port electrically actuated selection valve (Valco VICI Cheminert C25-3180D 06B–0699C, Houston, TX, USA) with a polytetrafluoroethylene (PTFE) tubing. PTFE tubing (0.8 mm i.d., Omnifit, Cambridge, UK) connected all the components of the SIA system. The syringe pump and the selection valve were controlled by AutoAnalysis Station 5.0 computer software (Sciware, Balearic Islands, Spain). As detection system consisted of an Ocean Optics (Orlando, FL, USA) USB 4000 charged coupled device detector (CCD) equipped with a pair of 600 mm optical cables, a Mikropack DH-2000-BAL deuterium halogen light source and an Ultem^®^ flow cell (SMA-Z-50 cell) with 50-mm optical path (130 µL inner volume).

### 3.5. Flow Manifold and Procedure

The sequence of steps for the determination of copper(II) and zinc(II) is shown in Table 4. It was divided in two parts, one corresponding to the Zn(II) determination using solid-phase extraction for the removal of Cu(II) and the other to the Cu(II) determination using PIM-based extraction to retain Zn(II). For the Zn(II) determination (Steps A–D, Table 4), reagent (Port 1, Figure 2), buffer (Port 2) and sample/standard (Port 5) were sequentially aspirated into the holding coil (HC). The sample/standard was aspirated through the Chelex 100 column (C2) via Port 5. Then, the staked zones were propelled to the flow cell (FC) where the absorbance was continuously monitored at 490 nm corresponding to the absorption maximum of the Zn(II)-PAR colored complex. For the Cu(II) determination (Steps E–M, Table 4), sample/standard was aspirated to the holding coil via Port 3, and then propelled through the PIM column (C1) to eliminate possible Zn(II) interference (Steps E and F). After passing through the PIM column, the flow was reversed twice to promote Zn(II) retention. After this procedure, reagent (Port 1), buffer (Port 2) and the Zn(II) free sample/standard (Port 1) were aspirated into the holding coil (HC). Then, these three zones were propelled to the flow cell (FC) for absorbance measurement of the Cu(II)-PAR colored complex at 490 nm.

At the end of the cycle, the PIM column was washed and reconditioned with 0.5 mol L^−1^ nitric acid (Port R3) and ultrapure water (carrier solution), sequentially.

Each absorbance value was calculated as the difference between the absorbances at 490 nm (wavelength of maximum absorption) and 800 nm; this subtraction aimed at minimizing the schlieren effect [24]. 

### 3.6. Sample Collection and Preparation

#### 3.6.1. Water Samples

River water samples (S1–S9) from various locations in the Porto area were collected, filtered (Acrodisc 25 mm syringe filters, 0.45 µm, Pall, USA) and acidified to pH 2 with nitric acid, according to the reference procedure [20]. Samples were kept refrigerated at 4 °C until analysis.

When copper(II) concentration was above 40 µg L^−1^, the water samples were diluted in order to fit the respective linear working range. In some samples, the zinc(II) concentration was below the detection limit, in which case the samples were spiked with zinc(II) (samples S1–S3).

#### 3.6.2. Soil Leachates Samples

The soil samples were collected in the northwest of Portugal using an acrylic cylinder that was pushed into the ground to collect a superficial soil core (about 20 cm depth). Two soil cores were collected. A commercial fertilizer was applied to the soil columns following the manufactured instructions.

To simulate the soil leaching process naturally occurring in the field, rain simulations were performed by passing 50 mL of previously collected rainwater (pH ≈6.6; conductivity ≈8.4 µS cm^−1^) through each soil core. Both the collected rainwater and the obtained leachates were filtered (Acrodisc 25 mm syringe filters, 0.45 µm, Pall, New York, NY, USA) and acidified to pH 2 with nitric acid.

Several simulations of rain were performed (for 5 consecutive days) and the soil leachates were collected. Samples were kept refrigerated until analysis. The soil leachates samples (S9–S14) were diluted when zinc(II) concentrations were above 40 µg L^−1^, in order to fit in the linear range of the zinc(II) calibration.

### 3.7. Reference Procedure

For validation purposes, the determinations of Cu(II) and Zn(II) in soil leachates and natural waters were carried out by inductively coupled plasma–optical emission spectrometry, ICP-OES (Perkin Elmer Optima 7000 dv, USA), and the results were compared with those obtained with the newly developed SIA method.

Additionally, the SIA system developed for the quantification of Cu(II) and Zn(II) was applied to the analysis of a certified water sample, ERM-CA011 (hard drinking water, LGC, Teddington, UK). The certified water sample was diluted in a multi-step fashion so that its concentration would fit within the linear range of the corresponding calibration curves.

## 4. Conclusions

With the aim of directly and individually quantifying copper(II) and zinc(II) with the same color reagent, two polymeric materials, namely a PIM and Chelex 100, packed in columns, were efficiently used. To the authors best knowledge, this was the first time that a PIM was used with this objective of on-line retaining and eliminating interferences from a sample. PVC-based PIMs containing D2EHPA proved to be efficient in retaining zinc(II), allowing for the quantification of copper(II). Chelex 100 was the polymeric material used to retain copper(II) at pH 2.0, as has already been reported in previous studies [9]. Zinc(II) was not retained at this pH, thus allowing for its quantification to be performed free of copper(II) interference.

Some flow-based methodologies have been previously developed [9,13,14,16,17,18,19] for the determination of these two metals in water samples using various chromogenic reagents (Table 5). Unlike these methods, the newly developed SIA system can determine copper(II) and zinc(II) individually and directly using a single manifold with the use of a single colorimetric reaction (PAR-Metal), thus reducing the time required per analysis while offering similar and in some cases better sensitivity. For example, the Zincon–Metal approach does not allow individual determination of the two metal ions [14,16,17,18].

The use of PAR as chromogenic reagent, instead of other reagents used by other authors (e.g., dithizone and PAN) [9,13], displays some advantages: PAR is considered a non-hazardous substance, unlike dithizone that is considered an eye and skin irritant according to European regulations (EC) [25]. PAR is also a water-soluble reagent and thus does not need the use of organic solvents in its preparation.

Overall, with the developed flow system, it is possible to attain in about 10 min the determination of copper(II) and zinc(II) individual content in natural waters, involving relatively low cost and portable equipment and minimizing the consumption of reagents.

## Figures and Tables

**Figure 1 molecules-25-05062-f001:**
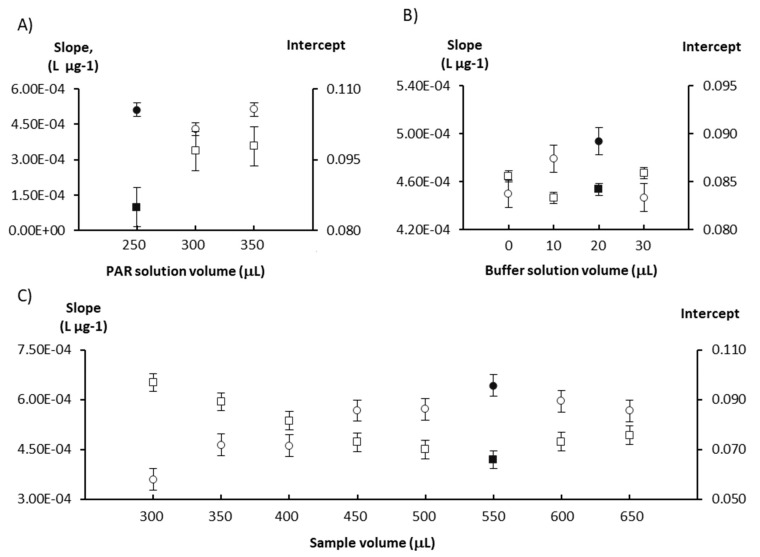
Study of the influence of the reagents (**A**,**B**) and sample (**C**) volumes on sensitivity expressed as the calibration curve slope (circles) and on the calibration curve intercept (squares). The chosen values are represented by solid-filled markers and the error bars represent the standard error.

**Figure 2 molecules-25-05062-f002:**
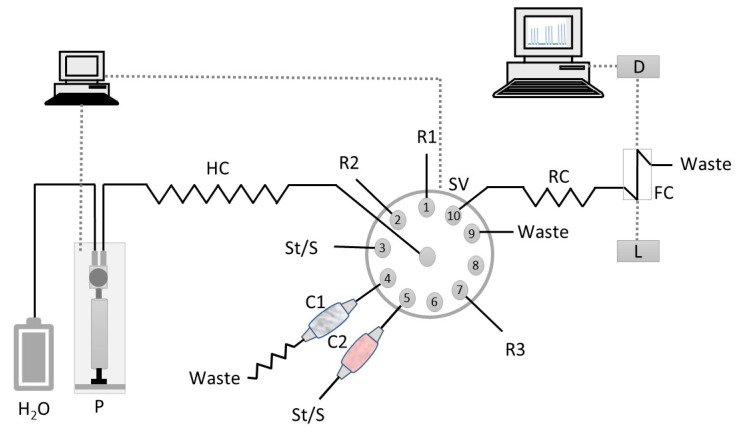
Flow manifold for Cu(II) and Zn(II) determination in waters and soil leachates. St/S, standard solution or sample; R1, PAR reagent (25 µmol L^−1^); R2, boric acid buffer solution (pH 11); R3– nitric acid solution (0.5 mol L^−1^); C1, PIM column; C2, Chelex 100 resin column; P, syringe pump; SV, selection valve; HC, holding coil (300 cm); RC, reaction coil (10 cm); D, CCD detector; L, light source; FC, Z flow cell (50-mm path length); W, waste.

**Table 1 molecules-25-05062-t001:** Interference study of metal ions ([M^n+^]) commonly present in environmental waters at their maximum expected concentrations ([M^n+^]_max_) [20]. SD, Standard deviation (n = 3).

Tested Ion	[M^n+^]_max_ in Streams, µg L^−1^	Tested [M^n+^], µg L^−1^	Interference in Cu(II) Determination, %	SD	Interference in Zn(II) Determination, %	SD
Al^3+^	400	400	−1.0	1.0	−1.3	0.9
Ca^2+^	15000	15000	1.0	1.0	−3.6	2.0
Co^2+^	0.2	10	1.6	0.1	1.5	0.1
Cu^2+^	<12	40	-	-	2.5	0.7
Fe^3+^	700	400 200	14.9 8.2	2.8 1.3	8.5 -	0.3 -
Mg^2+^	4000	5000 2500	3.1 -	2.6 -	35.8 −4.3	5.8 3.0
Mn^2+^	7	50	4.3	3.0	1.4	0.3
Ni^2+^	1	50	4.5	1.8	1.2	0.4
Zn^2+^	20	40	5.1	2.1	-	-

**Table 2 molecules-25-05062-t002:** Calibration curves and dynamic concentration ranges for copper(II) and zinc(II) and their respective limits of detection (LOD). A, absorbance; SD, standard deviation; M^2+^, metal ion.

Metal Ion	Dynamic Range (µg L^−1^)	Calibration Curve ^a^ A = (Slope ± SD) [M^2+^] + Intercept ± SD	LOD (µg L^−1^)	RSD (%)
Copper(II)	10.0–40.0	A = (9.00 × 10^−4^ ± 1.00 × 10^−4^) [Cu^2+^] + 0.112 ± 0.007	3.1	2.0
Zinc(II)	10.0–40.0	A = (1.80 × 10^−3^ ± 1.0 × 10^−4^) [Zn^2+^] + 0.099 ± 0.003	5.6	1.3

RSD, relative standard deviation; ^a^ n = 3.

**Table 3 molecules-25-05062-t003:** Comparison of the results obtained with the newly developed SIA system for copper(II) and zinc(II) determination (three replicates) with those obtained with ICP-OES (two replicates). S1–S9, river water samples; S10–S14, soil leachate samples; SD, standard deviation; RD, Relative deviation.

Sample ID	Copper(II)					Zinc(II)				
	SIA		ICP			SIA	ICP			
	[Cu^2+^] µg L^−1^	SD	[Cu^2+^] µg L^−1^	SD	RD %	[Zn^2+^] µg L^−1^	SD	[Zn^2+^] µg L^−1^	SD	RD %
S1 *	14.9	1.2	14.3	0.1	+3.7	22.8	2.4	21.6	0.3	+5.4
S2 *	19.5	2.6	20.6	0.3	−5.8	20.1	0.9	19.3	0.3	+3.9
S3 *	22.8	1.8	20.8	0.3	+6.4	31.4	3.0	33.5	0.4	−6.5
S4	197	8	183	3	+7.6	<LOD	-	<LOD	-	-
S5	140	8	143	2	−2.3	<LOD	-	<LOD	-	-
S6	107	5	114	2	−6.4	<LOD	-	<LOD	-	-
S7	72.2	4.1	74	3	−2.4	<LOD	-	<LOD	-	-
S8	156	10	143	5	+9.3	<LOD	-	<LOD	-	-
S9	<LOD	-	<LOD	-	-	<LOD	-	<LOD	-	-
S10	<LOD	-	2.08	0.1	-	15.2	2.0	14.9	0.2	+2.0
S11	<LOD	-	2.71	0.1	-	14.3	1.6	14.6	0.3	−2.0
S12	<LOD	-	<LOD	-	-	340	12	332	3	+2.5
S13	<LOD	-	<LOD	-	-	32.9	3.5	33.2	0.3	−0.9
S14	<LOD	-	0.58	0.1	-	203	3	193	3	+4.9

* Samples spiked with zinc(II).

**Table 4 molecules-25-05062-t004:** Experimental protocol for the copper(II) and zinc(II) determination.

Step	Selection Valve Position	Volume (mL)	Flow-Rate (mL min^−1^)	Description
Preliminary steps before starting consecutive cycles		5.000	-	Syringe reset position—syringe filled with carrier
		1.000	5.000	Propel carrier (water) to waste
A	1	0.250	3.529	Aspirate PAR solution
B	2	0.020	3.529	Aspirate boric acid buffer solution
C	5	0.550	2.000	Aspirate standard/sample through the Chelex 100 column to eliminate Cu(II) interference
D	10	2.100	3.529	Propel through the spectrometer flow cell for Zn(II) quantification
				Fill the syringe with carrier
E	3	0.550	2.000	Aspirate standard/sample
F	4	0.600	2.000	Propel through the PIM column to eliminate Zn(II) interference by retaining Zn(II)
G	4	0.250	2.000	Aspirate standard/sample through the PIM column to promote retention of Zn(II)
H	4	0.250	2.000	Propel standard/sample through the PIM column to promote retention of Zn(II)
I	9	0.250	3.529	Dispense to waste the left residues in the holding coil
J	1	0.250	3.529	Aspirate PAR solution
K	2	0.020	3.529	Aspirate boric acid buffer solution
L	4	0.550	2.000	Aspirate Zn(II) free standard/sample solution
M	10	2.100	3.529	Propel through the spectrometer flow cell for Cu(II) quantification
				Fill the syringe with carrier
R	7	0.500	5.000	Aspirate HNO_3_ solution
S	4	1.500	5.000	Propel through the PIM column—cleaning step

**Table 5 molecules-25-05062-t005:** Analytical features of flow-based systems developed for copper(II) and zinc(II) spectrophotometric determination in water samples (presented in descending chronological order).

System	Sample	Sample Volume (µL)	SPE	Reagent	Sample Throughput (h^−1^)	LOD (µg L^−1^)	Ref.
SIA	Natural waters	550	PIM and Chelex 100	PAR	6	Cu, 3.1 Zn, 5.6	This work
SIA	Water and soil leachates	413	Chelex 100	PAN	3	Cu, 3.0 Zn, 1.4	[9]
µSI-LOV	Freshwaters	600	NTA	Dithizone	Cu, 15 Zn, 13	Cu, 0.11 Zn, 2.39	[13]
SIC	Water	90	-	PAR	9	Cu, 13 Zn, 13	[19]
MSFIA	Waters	400	-	Zincon	43	Cu, 0.1 Zn, 2	[14]
SIA	Water samples	150	-	Zincon	36	Cu, 48 Zn, 13	[17]
BIS-FIA	Waters, pharmaceuticals and soils	1000	Sephadex QAE A-25	Zincon	15	Cu, 29 Zn, 40	[16]
FIA	Water and brass		Chelex 100	Zincon	70	Cu, 800 Zn, 350	[18]

SPE, solid phase extraction; LOD, limit of detection; Ref., Reference; SIA, sequential injection analysis; PIM, polymer inclusion membrane; PAR, 4-(2-pyridylazo)resorcinol; PAN, 1-(2-pyridilazo)-2-naphtol; µSI-LOV, micro sequential injection-lab-on-valve; SIC, sequential injection chromatography; MSFIA, multi-syringe injection analysis; BIS-FIA, bead injection spectrometry-flow injection analysis; FIA, flow injection analysis.

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
