# Peer review of "Use of a Polymer Inclusion Membrane and a Chelating Resin for the Flow-Based Sequential Determination of Copper(II) and Zinc(II) in Natural Waters and Soil Leachates"

_molecules, 2020, doi:10.3390/molecules25215062_

Round 1

Reviewer 1 Report

This manuscript describes the the flow-based sequential determination of copper (II) and zinc (II) in natural water and soil leachate using polymer inclusion membrane and a chelating resin. The following revisions are provided for the authors’ consideration.

  1. The pH of the natural water and soil leachate is approximately 6-9. Please explain the reasons for choosing a buffer solution with pH=11.
  2. In the interference experiment, Mg2+ interferences greatly, which limits the practical application of this device. Can the authors comment on this?
  3. Compared with other works, the performance of this device (dynamic range, relative deviation, etc.) is not outstanding, the innovation and advantages of this design need to be further elaborated by the authors.
  4. The stability and durability of this device should be supplemented.

Reviewer 2 Report

In my opinion, the reviewed work is an example of a manuscript describing a properly planned experiment in an efficient and clear manner. I have no objections to both the scientific and technical side of the work. Apart from minor linguistic errors, I do not find any shortcomings, thus I recommend the manuscript for publication in an unchanged form.

Author Response

This reviewer did not require any changes or comments.

Reviewer 3 Report

The concept of the manuscript is poor, I am not convinced by their unnecessary complicated system nor by the claimed selectivity.

The authors describe a SIA system for detection of Co and Zn using 4-(2-Pyridylazo) resorcinol (PAR) color reagent with claimed very good selectivity. They have employed two separate columns to discriminate between Zn and Co. The authors have reported an unnecessary complication of a previous work (ref 9) and some theoretical justifications are mandatory.

PAR is a reagent that react with a large number of heavy metal such as Cr (III), Mn (II), Fe (III), Co (II), Ni (II), Cu (II), Zn (II) and Cd (II) characterized by https://doi.org/10.1002/aoc.5528 or used for extraction of Cd(II), Co(II), Cu(II), Mn(II), Ni(IV), Pb(II), and Zn(II) as reported by https://doi.org/10.1016/j.talanta.2019.05.041. The authors must thoroughly justify with theoretical reasons why and how they observed reaction of PAR only with Zn and Co.

The authors have made an unnecessary complicated system using two columns to retain Co or Zn in order to detect only one analyte. Also, a simpler alternative already published by the authors (ref 9) is to use only one column to retain only one analyte and detect the other and the sum of both analytes. The Chelex 100 usage to retain Co was thus already reported and the second polymer seems to be rather inefficient since multiple passages are require through that column.

In fact any columns usage seems entirely unnecessary, both Co and Zn could be detected from the same sample without separation by recording the entire spectra and principal component analysis see https://doi.org/10.1016/j.ab.2019.03.007.

The authors use a strange and inappropriate method to extract the water-soluble fraction of Co and Zn from soil: passage of rainwater. Firstly, in agrochemical studies the interest is in the fraction available for plant uptake and for standardization specific buffers are used to compare the results from different soils. Secondly, rain water composition is variable even in the same location (e.g. due to episodes of pollution, dry/wet season) and no reproducible results can be obtained.

Round 2

Reviewer 1 Report

The manuscript can be accepted in present form

Author Response

No comment or change was asked.

Reviewer 3 Report

Dear Editor,

I am not convinced by the response of the authors, they have used an unnecesary complicated system and a non-specific reagent to acheieve a certain degree of selectivity (unmatched to AAS or even electrochemical methods). Also the usage of rain water is completely inapropriate even for environmental studies, e.g. how they know that the water was not polluted with heavy metals? At minimum, a comparison with purrified water must be done (again the rain water is just a complication to make a complicated/beautiful manuscript).

Best regards
